# Repurposing Mitomycin C in Combination with Pentamidine or Gentamicin to Treat Infections with Multi-Drug-Resistant (MDR) *Pseudomonas aeruginosa*

**DOI:** 10.3390/antibiotics13020177

**Published:** 2024-02-10

**Authors:** Elin Svedholm, Benjamin Bruce, Benjamin J. Parcell, Peter J. Coote

**Affiliations:** 1Biomedical Sciences Research Complex, School of Biology, University of St Andrews, The North Haugh, St Andrews, Fife KY16 9ST, UK; elin.svedholm@imbim.uu.se (E.S.); bb200@st-andrews.ac.uk (B.B.); 2NHS Tayside, Medical Microbiology, Ninewells Hospital and Medical School, Dundee DD1 9SY, UK; benjamin.parcell@nhs.scot

**Keywords:** *Galleria mellonella*, drug repurposing, antibiotic resistance, antibacterial, synergy, antibiotic resistance breaker, MexAB-OprM, combination therapy

## Abstract

The aims of this study were (i) to determine if the combination of mitomycin C with pentamidine or existing antibiotics resulted in enhanced efficacy versus infections with MDR *P. aeruginosa* in vivo; and (ii) to determine if the doses of mitomycin C and pentamidine in combination can be reduced to levels that are non-toxic in humans but still retain antibacterial activity. Resistant clinical isolates of *P. aeruginosa*, a mutant strain over-expressing the MexAB-OprM resistance nodulation division (RND) efflux pump and a strain with three RND pumps deleted, were used. MIC assays indicated that all strains were sensitive to mitomycin C, but deletion of three RND pumps resulted in hypersensitivity and over-expression of MexAB-OprM caused some resistance. These results imply that mitomycin C is a substrate of the RND efflux pumps. Mitomycin C monotherapy successfully treated infected *Galleria mellonella* larvae, albeit at doses too high for human administration. Checkerboard and time–kill assays showed that the combination of mitomycin C with pentamidine, or the antibiotic gentamicin, resulted in synergistic inhibition of most *P. aeruginosa* strains in vitro. In vivo, administration of a combination therapy of mitomycin C with pentamidine, or gentamicin, to *G. mellonella* larvae infected with *P. aeruginosa* resulted in enhanced efficacy compared with monotherapies for the majority of MDR clinical isolates. Notably, the therapeutic benefit conferred by the combination therapy occurred with doses of mitomycin C close to those used in human medicine. Thus, repurposing mitomycin C in combination therapies to target MDR *P. aeruginosa* infections merits further investigation.

## 1. Introduction

*Pseudomonas aeruginosa* is an opportunistic, nosocomial Gram-negative pathogen that has become an increasing global threat due to its rapid spread and the emergence of multi-drug-resistant (MDR) strains (defined as being resistant to three or more different classes of antibiotics). The increasing spread of MDR *P. aeruginosa* strains poses a particular threat since they are often associated with hospital outbreaks where clinically vulnerable patients are infected. This can result in a range of conditions, including hospital-acquired and ventilator-associated pneumonia, which are becoming increasingly difficult to treat due to the resistance observed [1]. *P. aeruginosa* causes 20.7% of all Gram-negative nosocomial pneumonia infections in the USA [2]. Furthermore, 25% of nosocomial *P. aeruginosa* infections in Thailand were due to MDR strains that resulted in higher mortality compared to their non-MDR counterparts [3]. Compounding these problems is the lack of new drugs in the pipeline that could target MDR *P. aeruginosa*, meaning that there is an urgent need to develop new treatments. However, developing new antibiotics is expensive and time-consuming with a high risk of failure. Repurposing existing drugs offers a potential alternative, since development time and costs are reduced, and they have already undergone clinical trials for safe use in humans [4]. Repurposing drugs as antimicrobials, or as adjuvants to boost the inhibitory effect of existing antimicrobials, has shown promise (reviewed in [5]). Such combination therapies, where two or more compounds work together with different mechanisms, ideally resulting in synergistic inhibition, can result in improved antimicrobial activity [6].

Repurposing of anticancer drugs, either alone or in combination, against MDR bacteria could represent a novel approach. There are some similarities between cancer cells and bacteria, such as their rapid proliferation, high metabolic rates, the ease of resistance development to therapeutic agents, and their ability to spread to other tissues [7,8]. Also, certain anticancer drugs—such as mitomycin C, bleomycin, and cisplatin—have been shown to have antimicrobial properties. Mitomycin C has a bactericidal effect against a range of bacteria, including *Acinetobacter baumannii*, *Staphylococcus aureus*, *Escherichia coli,* and *P. aeruginosa* [9,10]. Treatment with mitomycin C increased the survival of *A. baumannii*-infected *G. mellonella* larvae [9] and Enterohemorrhagic *Escherichia coli*-infected *Caenorhabditis elegans* [10]. Mitomycin C is an anticancer drug that inhibits DNA synthesis by crosslinking the strands in double-stranded DNA after activation through the reduction of the quinone group of the molecule and subsequent alkylation of DNA bases [11]. Although effective, mitomycin C is toxic, with a risk of cumulative myelosuppression and haemolytic uraemic syndrome [12]. Consequently, mitomycin C is a candidate for the development of novel combination therapies against MDR *P. aeruginosa* where, even at the low tolerable doses used in human therapy, the drug could act in synergy, or as a resistance breaker, when combined with existing antibiotics.

Another drug that has potential to be repurposed for use in combination with antibiotics is pentamidine. Pentamidine is an antiprotozoal treatment for leishmaniasis, trypanosomiasis, and Pneumocystis pneumonia, affecting RNA polymerase activity [13]. It shows antimicrobial activity but at concentrations that are not clinically possible in humans [14]. However, it has potential as an adjuvant in combination treatments. Stokes et al. [14] showed that pentamidine successfully sensitised Gram-negative pathogens to antibiotics typically reserved for treating Gram-positive infections and successfully sensitised colistin-resistant *A. baumannii* in a mouse model. The success of this synergy is due to the interactions of pentamidine with the lipopolysaccharide (LPS) of the Gram-negative outer membrane; pentamidine causes enhanced release of LPS from the outer membrane and compromises membrane integrity [15]. Wu et al. [6] incorporated pentamidine into an in vitro screen of 1374 FDA-approved non-antibiotic compounds with the aim of identifying any synergy that would be beneficial in treating Gram-negative bacterial infections with repurposed drugs. Pentamidine reduced the concentration of mitomycin C required for antimicrobial efficacy against several Gram-negative pathogens, including *P. aeruginosa*. However, in vivo testing was not undertaken to determine if the combination was effective at treating infection in a living system.

The aims of this study were (i) to determine if the combination of mitomycin C with pentamidine resulted in enhanced efficacy versus real infections with MDR *P. aeruginosa* in vivo using a *Galleria mellonella* infection model; (ii) to determine if the doses of mitomycin C and pentamidine in combination can be reduced to levels that are non-toxic in humans but still retain antibacterial activity; and (iii) to determine if the combination of mitomycin C with selected antibiotics can potentiate the inhibitory activity of each drug and result in enhanced efficacy in vivo.

## 2. Results

### 2.1. MDR Strains of P. aeruginosa Are Susceptible to the Anticancer Drug, Mitomycin C

The *P. aeruginosa* strains used in this study are shown in Table 1.

All the *P. aeruginosa* strains tested were inhibited by mitomycin C at concentrations comparable to antibiotic MICs (Table 2). The least susceptible strain was PAM1032, which over-expresses the MexAB-OprM resistance nodulation division (RND) efflux pump, and conversely, the most susceptible strain was PAM1626 with the triple deletion of three RND efflux pumps.

Triple deletion of the three RND pumps also conferred susceptibility to pentamidine because all the other strains tested were resistant to this drug. According to the European Committee on Antimicrobial Susceptibility Testing (EUCAST), sensitivity to ciprofloxacin is defined as ≤0.001 mg/L and resistance > 0.5 mg/L, and sensitivity to meropenem is ≤2 mg/L and resistance > 8 mg/L [18]. EUCAST does not define sensitivity or resistance to gentamicin for *P. aeruginosa* [18]. Exposure of two antibiotic-resistant clinical isolates (NCTC13437 and CR-BJP-POR) to these antibiotics revealed that NCTC13437 is resistant to both meropenem and ciprofloxacin and possessed a high MIC for gentamicin. In contrast, CR-BJP-POR was resistant to meropenem, displayed intermediate resistance to ciprofloxacin, and had a low MIC for gentamicin.

### 2.2. Treatment of G. mellonella Larvae Infected with Different P. aeruginosa Strains with Mitomycin C Results in Significant Therapeutic Benefit

The effect of treatment with single doses (administered 2 h post-infection (p.i)) of mitomycin C on *G. mellonella* larvae infected with a lethal dose of each of the *P. aeruginosa* strains is shown in Figure 1. Infected larvae were also treated with pentamidine monotherapy, but at the highest dose tested (100 mg/kg), no therapeutic benefit was observed. A small therapeutic effect of pentamidine at the highest dose tested (100 mg/kg) was only observed with larvae infected with PAM1626 where 20% of larvae survived 96 h p.i. The lack of efficacy of pentamidine monotherapy is supported by the very high MIC values that were observed for this compound (apart from that observed for PAM1626 (Table 2)).

Single, increasing doses of mitomycin C administered to larvae infected with each of the *P. aeruginosa* strains resulted in dose-dependent efficacy. The degree of efficacy conferred was also dependent on the individual strains of *P. aeruginosa* tested. For example, in larvae infected with strain PAM1032 (over-expressing the MexAB-OprM efflux pump), a dose of 50 mg/kg mitomycin C resulted in approximately 50% survival after 96 h post-infection (p.i), but in larvae infected with PAM1626 (three RND efflux pumps deleted), a dose of only 1.56 mg/kg conferred approximately 90% survival after 96 h p.i. This variable degree of efficacy conferred by mitomycin C correlated with the degree of in vitro sensitivity of these two strains to the drug (Table 2). Mitomycin C treatment of larvae infected with the other *P. aeruginosa* strains also resulted in high levels of therapeutic benefit—a single dose of 6.25 mg/kg resulted in nearly 100% survival 96 h p.i of larvae infected with *P. aeruginosa* CR-BJP-POR, but higher doses of 12.5 or 25 mg/kg were required to confer a similar degree of survival on larvae infected with strains NCTC13437, CR-BJP-VIM, or PAM1020.

The effect of therapy with mitomycin C on the internal burden of bacteria in larvae infected with two of the antibiotic-resistant strains of *P. aeruginosa* (NCTC13437 and CR-BJP-POR) is shown in Figure 2. After 24 h p.i, the mean numbers of infecting bacteria in larvae infected with either strain were significantly reduced in a dose-dependent fashion by mitomycin C compared to mock treatment with PBS. After 96 h p.i, bacterial numbers were reduced further after exposure to the two highest doses, and at the highest dose of mitomycin C tested, numbers were reduced such that no viable bacteria of either strain were recovered (the detection limit of the assay was 100 cfu/mL).

Together, these results show that monotherapy with mitomycin C is bactericidal versus infecting *P. aeruginosa* and results in significant therapeutic benefit against infections by this organism in vivo.

### 2.3. Combinations of Mitomycin C with Pentamidine, or Gentamicin, In Vitro Result in Synergistic, Bactericidal Inhibition of P. aeruginosa

Checkerboard assays showing the effect of different mitomycin C and pentamidine or gentamicin combinations on the growth of *P. aeruginosa* strains are shown in Figure 3. Synergistic inhibition was observed for the combination of mitomycin C with pentamidine for all the *P. aeruginosa* strains (Figure 3a). The strongest synergy identified was against NCTC10662 and PAM1032 (FICI—0.25) and the weakest was against NCTC13437 and PAM1626 (FICI—0.5). Furthermore, synergistic inhibition was also observed for the combination of mitomycin C with gentamicin for *P. aeruginosa* strain CR-BJP-POR (FICI—0.5), but only an additive effect was observed for the same combination against NCTC13437 (FICI—0.75) (Figure 3b). No synergy was detected between combinations of mitomycin C with meropenem or ciprofloxacin against either CR-BJP-POR or NCTC13437.

To determine if the synergistic inhibition observed in the checkerboard assays was bactericidal or bacteriostatic, time–kill assays were performed. The effect of exposure to PBS, pentamidine, gentamicin, or mitomycin C alone (at MIC_100_, MIC_50_, or MIC_25_) and in combination with mitomycin C on the viability of each *P. aeruginosa* strain is shown in Figure 4. Controls of each strain, mock-treated with PBS, increased in cell number over the 24 h duration of the experiment. Exposure to pentamidine alone resulted in a small decrease in the viability of all the *P. aeruginosa* strains after 2 h, but after 24 h, all strains recovered such that population viabilities were comparable with the PBS controls (Figure 4a). Exposure to mitomycin C alone also resulted in a small decrease in the viability of all strains after 2 h, but after 24 h, all recovered, but not to the full extent shown with the PBS-treated cells (Figure 4a,b). Notably, the combination of mitomycin C with pentamidine resulted in a large decline in viable numbers after 6 h of exposure with all strains except NCTC13437, where the decline in viable numbers was smaller and plateaued after just 4 h of exposure (Figure 4a). The loss of viability induced by the combination was greatest with NCTC10662, whereby no viable cells were recovered after 4 h (the detection limit of the assay was 100 colony-forming units (cfu)/mL) (Figure 4a).

Exposure of CR-BJP-POR and NCTC13437 to gentamicin or mitomycin C alone resulted in an initial loss of viability after 2 or 4 h for mitomycin C and gentamicin, respectively, followed by recovery of viable numbers after 96 h to levels slightly less than the PBS-treated controls (Figure 4b). Exposure to the combination of mitomycin C with gentamicin resulted in a large reduction in viable numbers for both strains tested, with no viable cells recovered after 4 or 6 h, for CR-BJP-POR and NCTC13437, respectively (Figure 4b). For CR-BJP-POR, no viable cells were recovered over the remaining duration of the experiment, but with NCTC13437, a minor recovery of viable numbers was detected with a low number of viable cells observed after 96 h (Figure 4b).

The American Society for Microbiology (ASM) definition of synergy with time–kill assays is a ≥2-log_10_ decrease in cfu/mL between the combination and its most active constituent after 24 h, and the number of surviving organisms in the presence of the combination must be ≥2 log_10_ cfu/mL below the starting inoculum (URL: https://journals.asm.org/abbreviations-conventions (accessed on 9/2/2024)). By this definition, the inhibition of *P. aeruginosa* by the combination of mitomycin C with pentamidine or gentamicin is synergistic. For most strains tested, none of the combinations, despite being potently bactericidal, eliminated all infecting *P. aeruginosa* bacteria over the duration of the experiment due to the detection of low numbers of surviving bacteria.

### 2.4. Combination Therapy with Mitomycin C and Pentamidine or Gentamicin of G. mellonella Larvae Infected with P. aeruginosa Results in Enhanced Efficacy Compared to Monotherapies

The effect of combination treatments compared with their constituent monotherapies is shown in Figure 5. Doses of each constituent drug in a combination that had minimal therapeutic benefit as a monotherapy were selected. This approach allowed the optimal identification of combinations that induced enhanced efficacy compared to the constituent monotherapies. A single dose of combination therapy at 2 h p.i with mitomycin C and pentamidine resulted in significantly enhanced efficacy compared to sham treatment with PBS or each monotherapy (Figure 5a). The therapeutic benefit conferred by combination therapy was observed for larvae infected with all the *P. aeruginosa* strains tested. Larvae infected with *P. aeruginosa* NCTC13437 were also treated with two doses of the combination and the monotherapies (at 2 and 4 h p.i) because the single-dose treatment only resulted in a small enhancement of survival (Figure 5a). Administration of two doses of the combination resulted in greatly improved therapeutic benefit compared with the single-dose treatment. This is consistent with the combination of mitomycin C and pentamidine having the smallest bactericidal effect on *P. aeruginosa* NCTC13437 in the in vitro time–kill assays compared with the other strains (Figure 4a).

The effect of combination therapy with mitomycin C and gentamicin compared with monotherapy is shown in Figure 5b. Administration of three doses of the combination (2, 4, and 6 h p.i) to larvae infected with *P. aeruginosa* CR-BJP-POR resulted in a large enhancement of survival compared with three doses of each monotherapy. Treatment with just one or two doses of this combination resulted in less significant improvement in larval survival. In contrast, administration of three doses of the same combination to larvae infected with *P. aeruginosa* NCTC13437 resulted in no enhanced efficacy at all (Figure 5b). The lack of evidence of enhanced efficacy by the combination of mitomycin C and gentamicin in vivo versus this strain is supported by the lack of synergy detected in vitro in the checkerboard assays (Figure 3b) and the apparent recovery of viable cells after 96 h exposure to the combination in the time–kill assays (Figure 4b). Nonetheless, the enhanced efficacy of the combination therapies observed in vivo is consistent with the inhibitory synergy between mitomycin C and pentamidine or gentamicin that was identified in vitro for most of the *P. aeruginosa* strains tested.

## 3. Discussion

In this work, mitomycin C showed inhibitory activity against a group of MDR and clinical isolates of *P. aeruginosa* with MIC values comparable with traditional antibiotics. Furthermore, the MIC values determined in this study compared favourably with those obtained for different *P. aeruginosa* strains in earlier studies [10,19]. Notably, mitomycin C was also shown to act as an effective antibiotic in vivo against infections by the same *P. aeruginosa* strains in *G. mellonella* larvae. This complements previous studies that showed mitomycin C was highly effective against *A. baumannii*-infected *G. mellonella* larvae [9] and Enterohemorrhagic *E. coli*-infected *Caenorhabditis elegans* [10]. Single doses of mitomycin C that conferred therapeutic benefit to infected larvae ranged from 6.25 mg/kg up to 50 mg/kg. In patients, a typical treatment dose for bladder cancer would be 20–40 mg of mitomycin C instilled into the bladder weekly [20]. This equates to a dose range of approximately 0.28–0.57 mg/kg weekly assuming a 70 kg patient. At these low dose concentrations, the data presented here show that the therapeutic benefit of mitomycin C therapy would be negligible. There would be little scope to increase the doses of mitomycin C that could be administered to patients due to the toxicity of the drug [12]. However, administration of mitomycin C in combination with other drugs could result in synergistic inhibition that would have the potential to reduce the dose of mitomycin C required to lower, less toxic levels, whilst still possessing significant antibacterial activity. In fact, this concept has been demonstrated in vitro, where the combination of mitomycin C with a tobramycin–ciprofloxacin hybrid antibiotic resulted in synergistic inhibition and significantly reduced the MIC of mitomycin C against MDR Gram-negative bacteria [19]. Similarly, Wu et al. [6] identified that the anti-protozoal drug pentamidine acted in synergy with mitomycin C against clinical isolates of Gram-negative bacteria, including *P. aeruginosa*. The authors showed that this combination had significantly enhanced efficacy against a colistin-resistant strain of *Enterobacter cloacae* compared with pentamidine or mitomycin C monotherapy in a *Caenorhabditis elegans* in vivo infection model. The data reported here further expand the potential of combination therapy with pentamidine and mitomycin C and also pentamidine with a common antibiotic, gentamicin, to treat MDR *P. aeruginosa* infections in vivo. Administration of combination therapies consisting of pentamidine with mitomycin C or gentamicin resulted in significantly enhanced efficacy against *G. mellonella* larvae infected with *P. aeruginosa* strains. Notably, the doses of mitomycin C used in these successful combination treatments were much lower than those required to provide similar levels of therapeutic benefit with mitomycin C monotherapy, for example, 0.78 mg/kg mitomycin C with gentamicin. This dose is close to the typical maximal dose of mitomycin C currently used in human therapy discussed previously and supports the concept of exploiting the antibacterial properties of mitomycin C in combination therapies for clinical application.

The MIC of mitomycin C was influenced by the status of the resistance nodulation division (RND) efflux pumps. The strain with three of these pumps deleted showed hypersensitivity to mitomycin C and the strain with over-expression of one of the RND pumps, MexAB-OprM, had the highest MIC of any of the strains tested. These data support a similar observation made by Domalaen et al. [19], who utilised different *P. aeruginosa* strains with similar altered RND efflux pump status, and implies a role for the RND efflux pumps in conferring resistance to mitomycin C. If RND efflux pumps do mediate the inhibitory effect of mitomycin C, this indicates that another potential combination therapy that could be explored to reduce the effective antibacterial concentration of mitomycin C, and thus toxicity, could involve the combination of the drug with an efflux pump inhibitor. That said, currently, there are no clinically approved efflux pump inhibitors available despite intensive research.

To conclude, this study has shown that mitomycin C monotherapy has potent anti-Pseudomonal activity both in vitro and in vivo but at concentrations/doses that are too high for use in human medicine due to the toxicity of the drug. However, the combination of pentamidine with mitomycin C or gentamicin results in synergistic, bactericidal killing of MDR strains of *P. aeruginosa* in vitro, and treatment with the same combination therapies results in enhanced efficacy in vivo against infections with the same strains in *G. mellonella* larvae. The efficacious combination treatments allow the administration of reduced doses of mitomycin C that are close to those used in human medicine, thus minimising toxicity whilst still conferring potent therapeutic benefit. Thus, repurposing the anticancer drug mitomycin C for use in combination with other approved drugs represents a potential route to develop new anti-Pseudomonal treatments and merits further investigation.

## 4. Materials and Methods

### 4.1. Bacteria and Growth Media

The *P. aeruginosa* strains used in this study are shown in Table 1. Strain PAO1 and the efflux pump mutants were gifted by Dr. Olga Lomovskaya, Qpex Biopharma, San Diego, CA, USA. *P. aeruginosa* NCTC10662 and 13437 were obtained from the National Collection of Type Cultures (NCTC) (http://www.phe-culturecollections.org.uk/collections/nctc.jsp). CR-BJP-POR is a clinical strain isolated from the sputum of a patient in intensive care with hospital-acquired pneumonia that did not respond to meropenem therapy. The strain is resistant to ceftazidime, imipenem, and piperacillin–tazobactam and displays intermediate resistance to meropenem and aztreonam. The isolate was positive for the modified carbapenemase inhibition test and the carbapenem inactivation method by the Scottish AMR Satellite Reference Laboratory, Glasgow, UK [21]. It was not found to possess any known carbapenemase enzymes at the Antimicrobial Resistance and Healthcare Associated Infections Reference Unit (AMRHAI), Public Health England, Colindale, and the antibiotic resistance profile was consistent with loss of the OprD porin and enhanced drug efflux. The strain CR-BJP-VIM is also a clinical isolate resistant to gentamicin, ciprofloxacin, piperacillin–tazobactam, ceftazidime, meropenem, and imipenem and was positive for a VIM (Verona Integron-Mediated Metallo-β-lactamase) gene. The strain was isolated from a leg ulcer swab. Both these clinical strains were provided by the co-author Dr Benjamin Parcell, NHS Tayside. All strains were cultured overnight in Mueller-Hinton Broth (MHB; Merck, Darmstadt, Germany) at 37 °C with shaking to prepare inocula for drug susceptibility testing in vitro and efficacy testing in vivo.

### 4.2. Reagents and G. mellonella Larvae

Ciprofloxacin (CIP), gentamicin (GEN), meropenem (MEM), and pentamidine (PEN) were purchased from Sigma–Aldrich Ltd. (Dorset, UK). Mitomycin C (MTC) was purchased from Tocris Bioscience (Bristol, UK). A 5 mL stock solution (10 mg/L) of ciprofloxacin (CIP) was made up in water with 100 µL of 1M HCl to fully dissolve. Concentrated stock solutions of gentamicin (GEN) and pentamidine (PEN) were prepared in sterile deionised water; meropenem (MEM) in water with 15% dimethyl sulphoxide DMSO (Fisher Scientific Ltd., Leicestershire, UK); and mitomycin C in 100% DMSO. Sub-stocks of all drugs for use in vitro and in vivo experiments were made up in sterile deionised water or MHB broth, respectively. *G. mellonella* larvae were obtained from UK Waxworms Ltd. (Sheffield, UK).

### 4.3. Antimicrobial Susceptibility and Checkerboard Assay

Minimum inhibitory concentrations (MICs) of MTC, PEN, and antibiotics against selected *P. aeruginosa* strains were determined in 96-well microplates as previously described [22]. The effect of combinations of MTC with PEN or gentamicin against selected *P. aeruginosa* strains was carried out by checkerboard assays in 96-well microplates. Briefly, plates were prepared by making doubling dilutions of MTC in MHB followed by subsequent addition of PEN or gentamicin. Wells were then inoculated with 1.0 × 10^6^ cfu/mL of *P. aeruginosa* cells and microplates incubated at 37 °C. After 24 h, each well was scored for visible growth and fractional inhibitory concentration index (FICI) values were calculated for each combination tested. The FICI value was calculated using the equation FICI = Ac/MICA + Bc/MICB, where Ac is the concentration of compound A when combined with compound B; MICA is the MIC of compound A alone; Bc is the concentration of compound B in combination with compound A; and MICB is the MIC of compound B alone. Synergy was defined at the point at which the FICI was ≤0.5. An additive effect was defined if the FICI was >1 and antagonism if it was ≥ 4 [23]. Selected *P. aeruginosa* strains were tested in duplicate.

### 4.4. Time–Kill Assay

*P. aeruginosa* cells (1.0 × 10^6^ cfu/mL) were exposed to PBS (control), MTC, and PEN or gentamicin alone or combinations of MTC and PEN or gentamicin in MHB at 37 °C. The drugs were used at concentrations that represented either MIC_50_ or MIC_25_. Viable bacteria were enumerated by serial dilution in MHB and plating on Nutrient Agar (NA) plates (Formedium Ltd., Hunstanton, England) after 2, 4, 6, and 24 h exposure. An initial inoculum was also enumerated as the starting cell number with no exposure to any treatments. Plates were incubated at 37 °C overnight prior to counting colonies. Susceptibility of *P. aeruginosa* strains was measured in duplicate and the mean ± standard error of the mean (SEM) was plotted.

### 4.5. G. mellonella Infection Model

Efficacy of MTC, GEN, and PEN alone or in combination versus *G. mellonella* larvae infected with *P. aeruginosa* strains was assessed as described previously [22]. Briefly, groups of 15 larvae were infected with an inoculum of 2.5 × 10^3^ cfu/mL of *P. aeruginosa* cells. Treatment with a single dose of drugs, or combinations of the same drugs, was administered 2 h post-infection (p.i). With some combinations, a second or third dose was given at 4 or 6 h p.i, respectively. All experiments were repeated in duplicate using larvae from a different batch and the data from these replicate experiments were pooled to give *n* = 30. Survival data were plotted using the Kaplan–Meier method [24] and comparisons were made between groups using the log-rank test [25]. In all comparisons with the negative control, it was the uninfected control (rather than the unmanipulated control) that was used and *p* ≤ 0.05 was considered significant.

The burden of *P. aeruginosa* cells within infected larvae treated with MTC was measured exactly as described previously [26,27]. Briefly, groups of 30 larvae were infected with *P. aeruginosa* cells (2.5 × 10^3^ cfu/mL) and MTC doses were administered at 2 h p.i. Larvae were incubated at 37 °C and after 24 h and 96 h p.i, five larvae were randomly sampled from each treatment group and surface decontaminated and anaesthetised by washing in absolute ethanol. Each larva was then placed in an Eppendorf tube containing 1 mL of sterile PBS and homogenised using a sterile pestle. Bacterial burden from individual caterpillars was then determined by serial dilution of the homogenate in MHB and plating on Pseudomonas Isolation Agar (Sigma–Aldrich Ltd., Dorset, UK). The detection limit for this assay was 100 cfu/mL of larval homogenate.

## Figures and Tables

**Figure 1 antibiotics-13-00177-f001:**
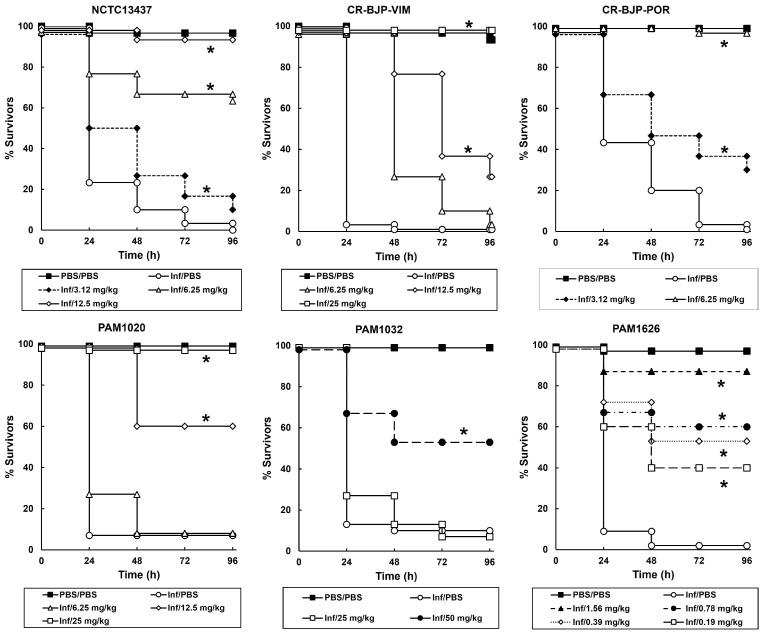
Mitomycin C monotherapy enhances the survival of *G. mellonella* larvae infected with a lethal dose (2.5 × 10^3^ cfu/mL) of *P. aeruginosa* strains. Infected larvae were treated with a single dose 2 h p.i of either PBS (mock ‘treated’) or increasing doses of mitomycin C, as indicated on the graph, and incubated at 37 °C. Surviving larvae were counted every 24 h for 96 h. The uninfected PBS/PBS group represents larvae sham-infected with sterile PBS and treated with sterile PBS. * Indicates significantly enhanced survival compared to infected larvae treated with PBS (*p* < 0.05, log-rank test with Holm correction for multiple comparisons); *n* = 30 (pooled from duplicate experiments).

**Figure 2 antibiotics-13-00177-f002:**
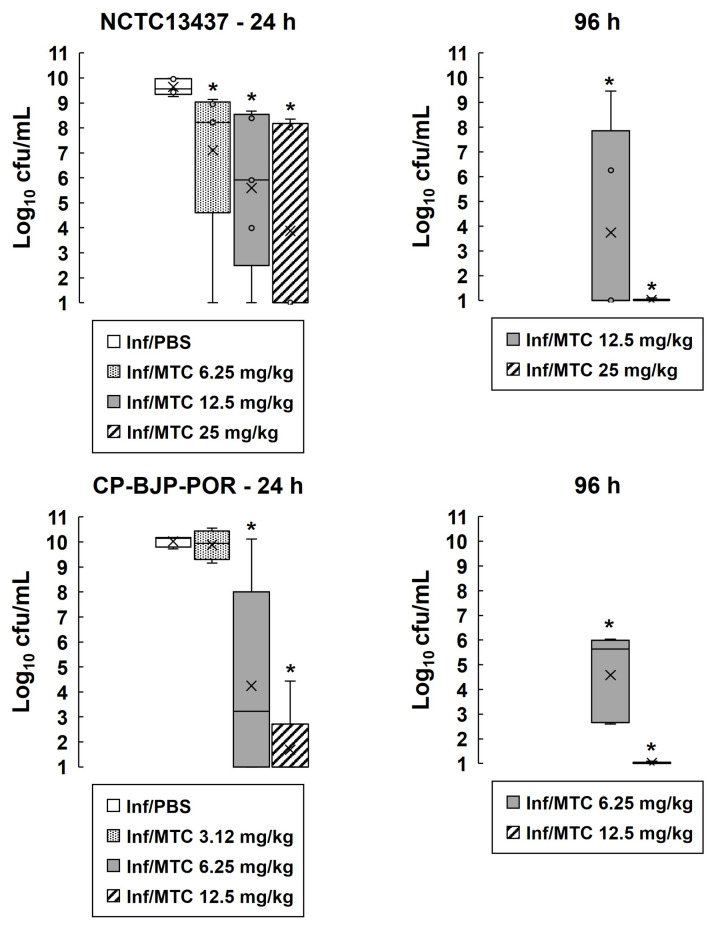
Mitomycin C monotherapy reduces the internal burden of *P. aeruginosa* strains in infected *G. mellonella* larvae. Larvae were infected with 2.5 × 10^3^ cfu/mL of either *P. aeruginosa* NCTC13437 or CP-BJP-POR and treated with either PBS (mock ‘treated’) or a single dose of mitomycin C at 2 h p.i (concentrations are shown on the graph). Larvae were incubated at 37 °C, and the internal burden of *P. aeruginosa* was determined from five individual larvae per treatment group after 24 and 96 h. The ‘x’ indicates the mean, the bar indicates the median, and the error bars show the highest and lowest values within the dataset. Outlier data are shown as independent points. The detection limit of the assay was log_10_ cfu/mL = 2 and larvae where no viable bacteria were recovered are plotted as log_10_ cfu/mL = 1. * Indicates doses of mitomycin C that conferred a significant reduction in bacterial burden compared to mock treatment with PBS (*p* < 0.05, the Mann–Whitney *U*-test; *n* = 5).

**Figure 3 antibiotics-13-00177-f003:**
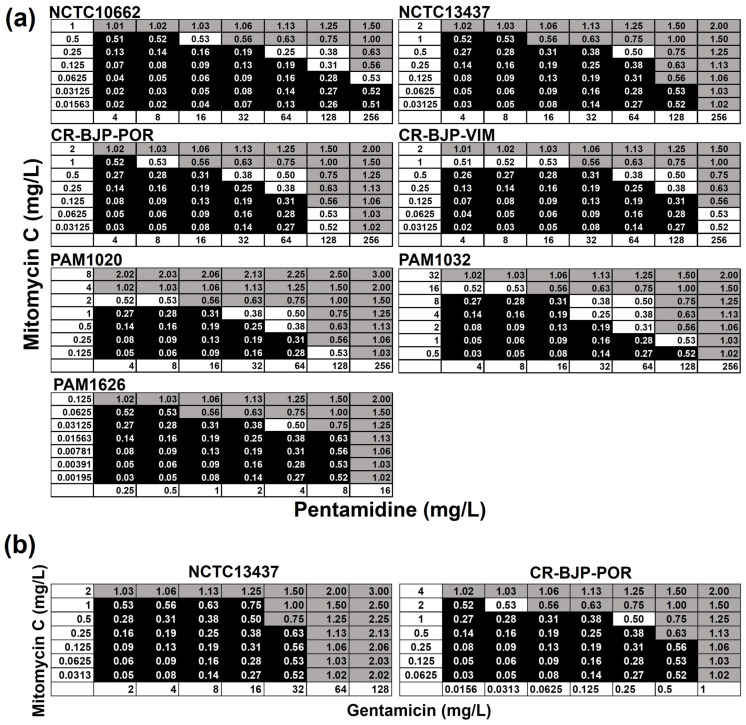
Checkerboard assays showing inhibition of *P. aeruginosa* growth by combinations of (**a**) mitomycin C with pentamidine or (**b**) mitomycin C with gentamicin. The fractional inhibitory concentration indices (FICIs) of each combination of mitomycin C with pentamidine, or gentamicin, were calculated versus each strain after 24 h at 37 °C and are shown in each square. Black squares indicate FICI values where bacterial growth occurred. Grey squares indicate wells where no growth occurred, but the FICI values were ≥0.5 (indicating inhibition was not synergistic). White squares also show where no growth occurred but where FICI values were 0.5 or less and thus indicate synergistic inhibition of growth. The experiments were performed in duplicate and representative results are shown.

**Figure 4 antibiotics-13-00177-f004:**
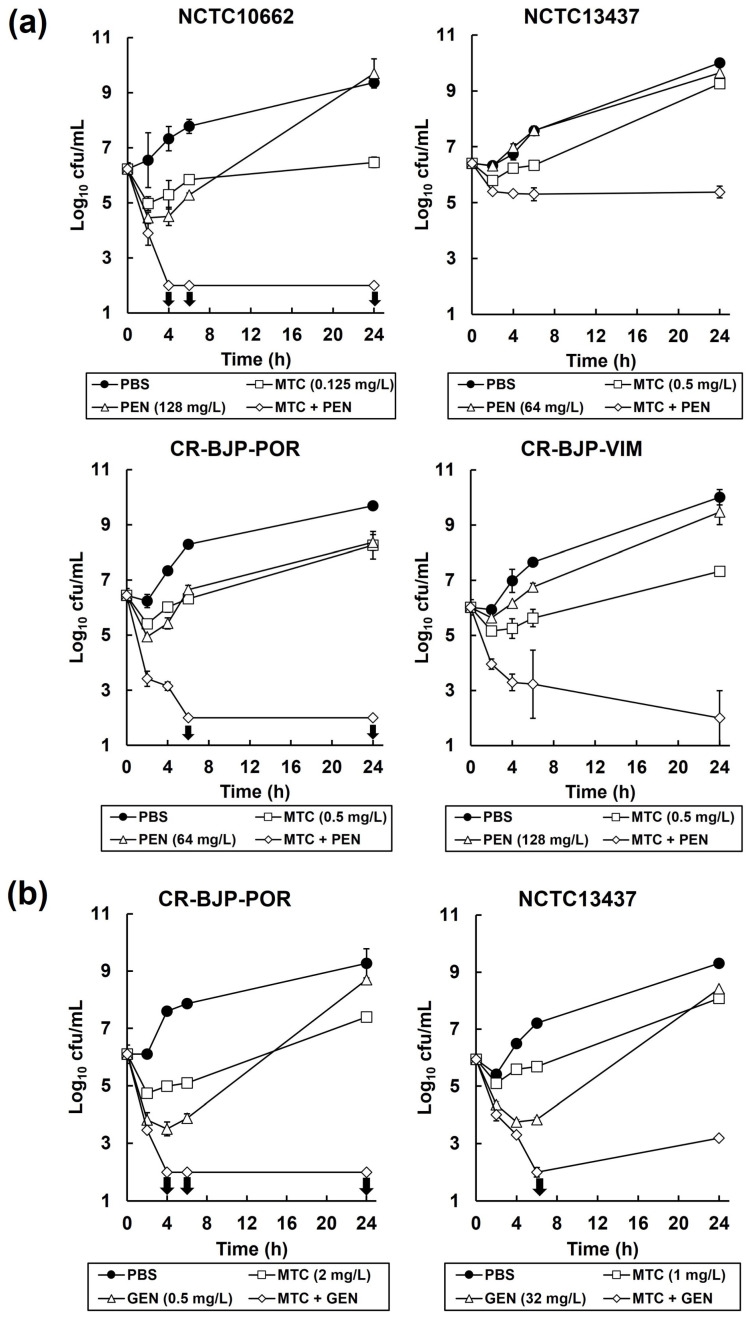
Time–kill assays showing the effect of (**a**) mitomycin C and pentamidine, or (**b**) mitomycin C and gentamicin, combinations on the viability of *P. aeruginosa*. Bacteria were exposed to mitomycin C, pentamidine, or gentamicin concentrations (shown on the graph) alone, or in combination, at either (**a**) MIC_0.25_ or (**b**) MIC_0.5_ for 24 h at 37 °C. Viable numbers were measured after 2, 4, 6, and 24 h exposure to each condition. Arrows indicate where no viable bacteria were detected (the detection limit of the assay was 100 cfu/mL). Each experiment was performed in duplicate and the mean ± SEM is shown.

**Figure 5 antibiotics-13-00177-f005:**
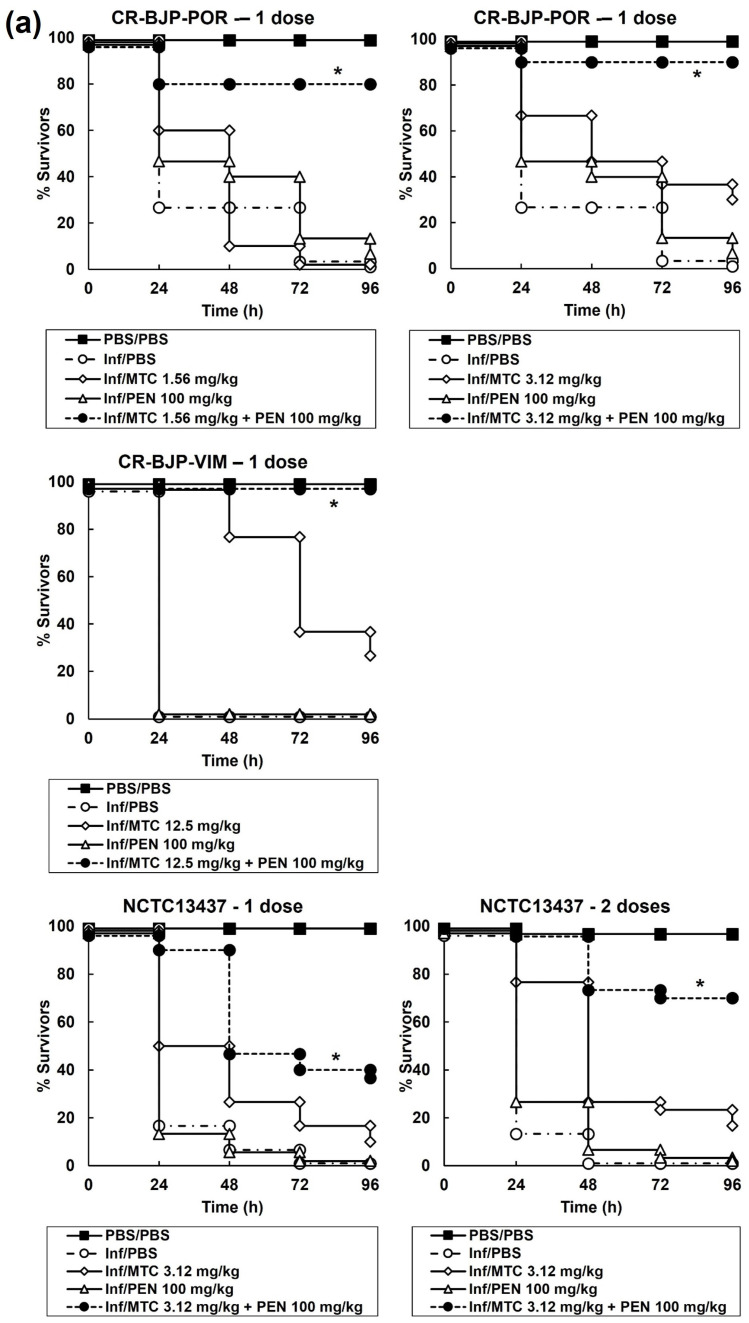
Effect of treatment with (**a**) mitomycin C or pentamidine monotherapy, or mitomycin C with pentamidine combinations, (**b**) mitomycin C or gentamicin monotherapy, or mitomycin C with gentamicin combinations on survival of *G. mellonella* larvae infected with 2.5 × 10^3^ cfu/mL of *P. aeruginosa* strains. Infected larvae were treated with PBS (mock ‘treated’), monotherapies, or mitomycin C combinations at the doses indicated on the graphs for each strain. Single, double, or triple doses of the treatments were administered 2, 4, or 6 h p.i, respectively, as indicated on the graphs. Larvae were incubated at 37 °C and surviving larvae were counted every 24 h for 96 h. The uninfected PBS/PBS group represents larvae sham-infected with sterile PBS and treated with sterile PBS. * Indicates significantly enhanced survival compared to each monotherapy alone (*p* < 0.05, log-rank test with Holm correction for multiple comparisons); *n* = 30 (pooled from duplicate experiments).

**Table 1 antibiotics-13-00177-t001:** *Pseudomonas aeruginosa* strains.

Strain	Genotype	Phenotype	Reference
NCTC10662	Clinical isolate	Antibiotic susceptible control strain	
NCTC13437	Clinical isolate producing VEB-1; VIM-10 β-lactamases	Resistant to β-lactams and fluoroquinolones by an unknown mechanism	[16]
CR-BJP-POR	Clinical isolate	Resistant to β-lactams via enhanced efflux or porin loss	Clinical isolate
CR-BJP-VIM	Clinical isolate producing a VIM β-lactamase	Resistant to β-lactams, aminoglycosides, fluoroquinolones	Clinical isolate
PAM1020	PA01 prototroph	Wild-type parent strain	[17]
PAM1032	*nalB*-type mutation	*mexAB*-*oprM* over-expressed	[17]
PAM1626	Δ*mexAB*-*oprM*::Cm; Δ*mexCD-oprJ*::Gm; Δ*mexEF-oprN*::ΩHg	*mexAB*-*oprM*; *mexCD-oprJ*; and *mexEF-oprN* deleted	[17]

**Table 2 antibiotics-13-00177-t002:** Antimicrobial minimum inhibitory concentrations (MICs) versus *Pseudomonas aeruginosa* strains. Each experiment was performed at least in duplicate. MTC—mitomycin C; PEN—pentamidine; GEN—gentamicin; MEM—meropenem; CIP—ciprofloxacin. - not tested.

Strain	Phenotype	MIC (mg/L)				
		MTC	PEN	GEN	MEM	CIP
NCTC10662	Antibiotic susceptible	0.5–1	512	-	-	-
NCTC13437	Resistant to β-lactams and fluoroquinolones	2	256	64–128	64–128	32
CR-BJP-POR	Resistant to β-lactams	2–4	256	1	8	0.25
CR-BJP-VIM	Resistant to β-lactams, aminoglycosides, fluoroquinolones	2	512	-	-	-
PAM1020	Isogenic parent strain of efflux pump mutants	4–8	256	-	-	-
PAM1032	Over-expression of MexAB-OprM	16–32	256	-	-	-
PAM1626	Triple deletion of MexAB-OprM, MexCD-OprJ, and MexEF-OprN	0.125	16	-	-	-

## Data Availability

Data can be made available by the corresponding author on request.

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
