# Peer review of "Repurposing Mitomycin C in Combination with Pentamidine or Gentamicin to Treat Infections with Multi-Drug-Resistant (MDR) *Pseudomonas aeruginosa"

_antibiotics, 2024, doi:10.3390/antibiotics13020177_

Round 1
Reviewer 1 Report
Comments and Suggestions for Authors
Svedholm et al. explored the effectiveness of combining mitomycin C with pentamidine or antibiotics like gentamicin against multi-drug resistant (MDR) P. aeruginosa infections. They performed In vitro and in vivo tests and reported that while mitomycin C alone is effective, its high dosage is too toxic for human use. They report combining it with pentamidine or gentamicin enhances its efficacy against MDR P. aeruginosa, allowing for lower, safer dosages. The authors suggest that repurposing mitomycin C in combination therapies could be a promising strategy for treating these infections, warranting further research.
The authors need to highlight the future implications of their research and address the limitations associated with the current study. For instance, performing the tests in triplicates, rather than duplicates, would have been more statistically reliable.
The selection of dosages in the study appears unclear. Specifically, the authors may need to provide a clear rationale for the dosages chosen in the experiments. For instance, those presented in Figure 4.
Comments on the Quality of English Language
The authors are requested to review the manuscript for grammatical and typographical errors and make the necessary corrections.
Author Response
Point 1. The implications of this research are stated in the final paragraph of the Discussion.
The in vitro experiments were performed in duplicate but the data generated was highly reproducible, for example the SEM values shown in the time-kill experiments (Figure 4a and b) show this. Furthermore, the G. mellonella experiments were also performed in duplicate with population sizes of n=30, but these were then pooled to make a population size of n=60. Again, evidence of the high reproducibility of this data is shown with the box plots in Figure 2. Thus, the duplicate experiments gave statistically reliable data.
Point 2. Doses were selected after pilot experiments assessed the therapeutic benefit of monotherapy (Figure 1). Subsequent dose selection for combinations (Figure 4) involved selecting doses of each constituent drug in the combination that had minimal therapeutic benefit alone. We have included an additional sentence line 241-244 for clarity.
Reviewer 2 Report
Comments and Suggestions for Authors
Please see attached file

Author Response
Point 1. We accept that using the term susceptibility is perhaps inappropriate for mitomycin C. Thus, we accept the reviewers point, and have reworded the sentence at line 101-102 reflect this.
Point 2. In the Methods section of the manuscript the precise methodology for infecting the larvae is described in reference [22]. We do not believe it is necessary to include an additional sentence.
Point 3. The reviewer makes a valid point regarding the dose of mitomycin C used to treat glaucoma. However, we cannot include this observation in the manuscript without the reviewer providing a publication reference for this dosage/treatment for glaucoma. Also, this treatment would represent a more topical-type administration of mitomycin C rather than i.v treatment that would be likely for a systemic infection. and, as such, is perhaps not an appropriate comparison to make.
Point 4. The authors make a valid point that mitomycin C induces the SOS response and thus could contribute to antibiotic resistance induction. However, we feel that this is outside the scope of this study because we are presenting data that identifies an effective treatment for MDR P. aeruginosa and have not studied induction of resistance.